# Neural Minimum Weight Perfect Matching for Quantum Error Codes

**Yotam Peled** [1]  **David Zenati** [1]  **Eliya Nachmani** [1]

## Abstract

Realizing the full potential of quantum computation requires Quantum Error Correction (QEC). QEC reduces error rates by encoding logical information across redundant physical qubits, enabling errors to be detected and corrected. A common decoder used for this task is Minimum Weight Perfect Matching (MWPM) a graph-based algorithm that relies on edge weights to identify the most likely error chains. In this work, we propose a data-driven decoder named Neural Minimum Weight Perfect Matching (NMWPM). Our decoder utilizes a hybrid architecture that integrates Graph Neural Networks (GNNs) to extract local syndrome features and Transformers to capture long-range global dependencies, which are then used to predict dynamic edge weights for the MWPM decoder. To facilitate training through the non-differentiable MWPM algorithm, we formulate a novel proxy loss function that enables end-to-end optimization. Our findings on the toric code under depolarizing noise demonstrate thresholds of 17.9% and 10.95%, nearing the 18.9% and 11.0% maximum likelihood bounds, highlighting the advantage of hybrid decoders that combine the predictive capabilities of neural networks with the algorithmic structure of classical matching.

## 1. Introduction

The realization of fault-tolerant quantum computing promises to unlock computational capabilities far exceeding the limits of classical algorithms (Steane, 1998; Ladd et al., 2010; Preskill, 2012). The theoretical foundation of quantum computing is built upon algorithms that outperform their classical counterparts. Examples such as Shor's factoring algorithm (Shor, 1994) and Grover's search (Grover,

1996) provide the mathematical proof of this advantage, signaling a paradigm shift for industries reliant on cryptography (Ekert, 1991; Bennett & Brassard, 2014), chemical simulation (Aspuru-Guzik et al., 2005), and complex optimization (Kadowaki & Nishimori, 1998; Bharti et al., 2022). Recent experimental milestones in quantum supremacy have further substantiated the transformative potential of quantum computing across a wide range of disciplines. (Arute et al., 2019; Huang et al., 2022; Madsen et al., 2022; Bluvstein et al., 2024; Bao et al., 2023). However, the fundamental unit of quantum information, the physical qubit, is inherently fragile. Susceptible to decoherence and operational errors arising from environmental interaction (Burnett et al., 2019; Etxezarreta Martinez et al., 2021) and imperfect control, physical qubits cannot sustain information long enough for complex calculations. QEC is therefore indispensable for bridging the gap between noisy physical hardware and reliable quantum computation. Among the various QEC schemes proposed (Panteleev & Kalachev, 2021; Breuckmann & Eberhardt, 2021), topological codes (Kitaev, 2003; Bombin & Martin-Delgado, 2006; Fowler et al., 2012; Chamberland et al., 2020) have emerged as a leading approach. In these architectures, logical information is encoded across a grid of physical qubits, allowing error detection via local measurements. The surface code utilizes an $L_{code} \times L_{code}$ grid of local interactions, where $L_{code}$ denotes the code distance. It is widely favored for its high error threshold, the critical noise level below which increasing the code size improves, rather than degrades, information protection. The efficacy of the surface code relies heavily on the performance of its decoder, the algorithm responsible for inferring errors from observed syndromes. The MWPM (Fowler, 2015) has established itself as the standard decoder for these codes. MWPM effectively casts the decoding task as a graph theory problem, seeking a perfect matching of minimum total weight on a graph where nodes correspond to syndrome defects and edges represent potential error chains. Despite its widespread adoption, standard MWPM simplifies the decoding problem by assuming independent error contributions from bit and phase flip, which restricts how correlations between faults can be incorporated. Thus, the decoder fails to exploit the rich statistical information in the specific syndrome distribution of each shot. Hybrid approaches that aim to parameterize classical decoders face a fundamental optimization barrier. The MWPM algorithm

[1]School of Electrical and Computer Engineering (ECE), Ben-Gurion University, Beer Sheva, Israel. Correspondence to: Yotam Peled <peledyot@post.bgu.ac.il>, Eliya Nachmani <eliyanac@bgu.ac.il>.

*Proceedings of the 43rd International Conference on Machine Learning*, Seoul, South Korea. PMLR 306, 2026. Copyright 2026 by the author(s).

relies on discrete, combinatorial operations that are inherently non-differentiable, as its output is a discrete binary assignment for every edge. This characteristic impedes standard backpropagation, effectively severing the gradient flow from the decoding decision back to the network parameters and making it difficult to learn optimal weighting strategies in an end-to-end fashion.

In this work, we address these challenges by introducing a novel, differentiable decoding framework that augments the classical MWPM algorithm with a hybrid deep learning architecture. Rather than replacing the matching algorithm, our approach empowers it; we employ a deep neural network to dynamically predict the optimal edge weights for the matching graph based on the observed syndrome. To achieve this, we introduce an architecture combining a GNN to capture local syndrome topology and Transformer (Vaswani et al., 2017) to model global dependencies. Crucially, we resolve the optimization challenge by formulating a proxy loss function, enabling gradient-based training of a network intended to drive a non-differentiable algorithm. Our specific contributions are as follows:

- **Novel Hybrid Architecture:** We propose a unified framework that combines a GNN and Transformer to predict dynamic edge weights from syndrome data. Our two stage architecture first employs a GNN to encode the local topology of the syndrome graph, followed by a global Transformer encoder that reasons about competing error chains across the entire lattice.

- **Ground Truth Generation:** We introduce an algorithmic procedure to generate labeled training data by reducing physical error configurations to valid matchings on the syndrome graph. This provides the supervised signal necessary for the network to learn to identify the error chains.

- **Differentiable Training Objective:** We enable the training of this algorithm by formulating the problem as edge classification, optimizing a binary cross-entropy (BCE) objective with an entropy regularization term. This differentiable proxy loss circumvents the inherent non differentiability of the MWPM algorithm.

- **Decoding Performance:** We evaluate our framework on the Toric Code (Kitaev, 1997) and the Rotated Surface Code (Bombín & Martin-Delgado, 2007) under depolarizing and independent noise models. Our results demonstrate that our neural augmented approach consistently outperforms standard MWPM baselines, achieving lower LER by effectively utilizing syndrome information that static priors ignore.

## 2. Related Work

Decoding quantum error-correcting codes is a complex and computationally intensive task (Kuo & Lu, 2020), which has driven the development of various approximate methods that prioritize computational efficiency over absolute optimality (Dennis et al., 2002; deMarti iOlius et al., 2024). Classical strategies for decoding typically frame the problem through graph-theoretic or probabilistic approaches, these include the union-find decoders which translate syndromes into graph problems (Delfosse & Nickerson, 2021); belief propagation, which is effective for sparse parity-check codes but is hindered by quantum degeneracy (Roffe et al., 2020; Wang & Tang, 2024); and tensor-network decoders that attain the highest accuracy at steep computational cost (Bravyi et al., 2014; goo, 2023). Another prominent approach is the MWPM, which reaches near-optimal thresholds under independent noise but suffers from poor scaling even with practical approximations (Fowler, 2015). While the MWPM decoder fundamentally relies on the blossom algorithm (Edmonds, 1965), its high computational cost in worst case scenarios has driven the development of faster implementations essential for real-time decoding (Higgott, 2022). Key optimized variants include the sparse blossom algorithm (Higgott & Gidney, 2025), and the linear-complexity Fusion Blossom (Wu & Zhong, 2023), which trade off between single thread efficiency and multi thread execution support. While these conventional methods are foundational, they possess limitations that restrict their practical utility in large-scale, fault-tolerant quantum systems (deMarti iOlius et al., 2024). Machine learning provides a powerful alternative, with diverse models that show superior speed and accuracy over traditional baselines while being uniquely suited to accommodate the correlated and device-specific noise that complicates classical decoding (Krenn et al., 2023; Wang & Tang, 2024; deMarti iOlius et al., 2024; Varsamopoulos et al., 2017; 2019; Harper et al., 2020; Magesan & Gambetta, 2020; Liu & Poulin, 2019; Maskara et al., 2019; Meinerz et al., 2022). Specifically, these architectures are employed in reinforcement learning, (Colomer et al., 2020; Sweke et al., 2020; Fitzek et al., 2020; Veeresh et al., 2024; Andreasson et al., 2019), GNN-based decoders (Lange et al., 2025), and transformer-based architectures (Choukroun & Wolf, 2024; Bausch et al., 2024; Zenati & Nachmani, 2025; Senior et al., 2025).

## 3. Background

### 3.1. Classical and Quantum Foundations

In classical error correction, redundancy is imposed on the logical information by embedding $k$ information bits into $n$ physical bits through a collection of parity constraints. These constraints are compactly represented by a binary parity-check matrix $H \in GF(2)^{(n-k) \times n}$ and the set of

valid codewords is given by:

$$\mathcal{C} = \{x \in GF(2)^n \mid Hx^T = 0\} \tag{1}$$

When an error $e$ occurs, it displaces the encoded word from $\mathcal{C}$, producing a nonzero syndrome $s = He^T$, which indicates violated parity constraints. Extending this construction to quantum information is nontrivial, as qubits are not classical binary variables, but two-level systems described by a state vector $|\psi\rangle$ that can exist in coherent superpositions:

$$|\psi\rangle = \alpha|0\rangle + \beta|1\rangle, \quad \text{where } \alpha, \beta \in \mathbb{C}, \quad |\alpha|^2 + |\beta|^2 = 1. \tag{2}$$

Here, $\alpha$ and $\beta$ are complex probability amplitudes satisfying the normalization condition, such that $|\alpha|^2$ and $|\beta|^2$ represent the probabilities of measuring the states $|0\rangle$ and $|1\rangle$, respectively. Since the no-cloning theorem prevents standard redundancy, errors are decomposed into the discrete Pauli basis $\{I, X, Y, Z\}$. representing identity, bit-flip, phase-flip, and combined flip operations, respectively. We define their transformations on an arbitrary state $|\psi\rangle$ as:

$$I|\psi\rangle = \alpha|0\rangle + \beta|1\rangle; \quad X|\psi\rangle = \alpha|1\rangle + \beta|0\rangle; \tag{3}$$

$$Y|\psi\rangle = i\alpha|1\rangle - i\beta|0\rangle; \quad Z|\psi\rangle = \alpha|0\rangle - \beta|1\rangle. \tag{4}$$

Under the standard Pauli channel model, an error type $k \in \{I, X, Y, Z\}$ occurs with probability $p_k$, satisfying the normalization condition $\sum_p p_k = 1$. Errors generalize to tensor products $E = P_1 \otimes \cdots \otimes P_n$ for $n$ qubits, representing a simultaneous operation where each $P_i$ acts on the $i$-th qubit. This results in a discrete but exponentially growing error space of size $4^n$ (compared to $2^n$ classically), a significant challenge for efficient error identification.

## 3.2. The Stabilizer Formalism

The stabilizer formalism (Gottesman, 1997) encodes quantum information by constraining an $n$-qubit system to the simultaneous $+1$ eigenspace of a commuting set of Pauli operators within the Hilbert space $\mathcal{H}_2^n$. These operators are drawn from the $n$-qubit Pauli group $\mathcal{P}_n$, which consists of tensor products of single-qubit Pauli matrices up to an overall phase. Errors manifest as operators that violate these symmetry constraints and can be identified through projective measurements that do not disturb the encoded logical information. Concretely, an $[[n, k, L_{\text{code}}]]$ stabilizer code is specified by choosing $m = n - k$ independent generators $\{S_i\}_{i=1}^m$ subject to the constraint that the group generated by these operators does not contain $-I$. The associated logical code space is given by

$$C_S = \{ |\psi\rangle \in \mathcal{H}_2^n \mid S_i|\psi\rangle = |\psi\rangle, \, \forall i \in \{1, \ldots, m\} \}. \tag{5}$$

Measuring the stabilizer generators produces a binary syndrome that reflects the commutation relations between an error operator and the stabilizers, thereby enabling error identification while preserving the logical state.

## 3.3. Degeneracy and Learning Motivation

A defining feature of quantum codes is *degeneracy*. Multiple distinct errors $E$ and $E'$ may produce identical syndromes. These errors are logically equivalent. Consequently, the decoding objective is not to identify the exact physical error, but to determine the correct equivalence class to which the error belongs. This phenomenon reframes decoding as a complex prediction task. Since the number of independent binary checks scales with the lattice area $L_{code}^2$, the syndrome space grows exponentially ($2^{O(L_{code}^2)}$ for surface codes), making the search for the optimal equivalence class computationally intensive. However, surface codes exhibit strong local geometric correlations and hierarchical error structures. These properties make the problem an ideal candidate for deep learning architectures.

## 3.4. Minimum Weight Perfect Matching

Let $G = (V, E)$ be a weighted undirected graph, where $V$ is a set of vertices and $E$ is a set of edges, with each edge $(u, v) \in E$ assigned a weight $w_{uv}$. A *matching* $M \subseteq E$ is a subset of edges such that no two edges share a common vertex. A matching is *perfect* if every vertex in $V$ is incident to exactly one edge in $M$. The MWPM problem seeks the perfect matching $M^*$ that minimizes the total weight,

$$M^* = \underset{M \in \mathcal{M}}{\arg\min} \sum_{(u,v) \in M} w_{uv} \tag{6}$$

where $\mathcal{M}$ denotes the set of all perfect matchings on $G$. The problem can be solved in polynomial time using the Blossom algorithm (Edmonds, 1965). The algorithm's defining feature is its handling of odd cycles, which typically block such searches. When an odd cycle is encountered, the algorithm contracts the loop into a single virtual node called a "blossom." This contraction simplifies the graph topology, allowing the algorithm to bypass the obstruction and find a global solution before expanding the blossom back to fix the local connections. In the context of topological decoding, the MWPM constructs a graph where vertices correspond to syndrome defects and edges represent potential error chains. The weight assigned to an edge functions as a probabilistic cost, typically derived from the negative log-likelihood of the error chain. Consequently, finding the minimum weight matching is equivalent to identifying the most probable physical error configuration consistent with the observed syndrome. Due to its polynomial efficiency and high accuracy, MWPM has become the standard decoder for surface codes.

## 4. Method

### 4.1. Framework Overview

We propose a hybrid framework leveraging geometric deep learning's inductive bias, specifically, its modeling of topological structure, alongside classical robustness. Rather than replacing MWPM, our approach augments it by substituting static edge weights with dynamic, learned probabilities derived from the syndrome. To enable this, we employ a supervised training where ground truth edge labels are derived from the underlying error configuration. The model is then optimized using a binary classification loss to predict the likelihood of each edge being part of the correction. The pipeline proceeds in three distinct stages:

(i) **Graph Construction and Preprocessing:** The syndrome measurement is mapped to a fully connected graph where nodes represent defects and edges represent potential error chains. We construct rich feature vectors for both nodes and edges, incorporating spatial coordinates, stabilizer types, and learned embeddings.

(ii) **Edge Weight Prediction:** We introduce the Quantum Weight Predictor (QWP), an architecture that processes the graph. We first employ a GNN backbone - specifically a TransformerConv (Shi et al., 2021) layer, to update node representations based on local topology. Subsequently, a Transformer Encoder processes the edges (formed by concatenating updated node pairs) to capture global dependencies. The network outputs a scalar probability $p_{ij}$ for every edge connecting nodes $i$ and $j$, representing the likelihood that the edge is part of the true error chain.

(iii) **Matching:** The network assigns an error probability $p_{ij}$ to every edge in the decoding graph. These probabilities are transformed into final edge weights $w_{ij}$ via the negative log-likelihood transform $w_{ij} = -\ln(p_{ij})$. These weights are fed into the standard MWPM algorithm to predict the final correction. The complete inference pipeline is formalized in the Algorithm 1. In this procedure, QWP($S$) executes the forward pass of our neural backbone, mapping the input syndrome $S \in \{0, 1\}^N$, where N is the total number of stabilizers, to a set of edge probabilities $\mathcal{D}$. A visual overview of this complete hybrid architecture, is presented in Figure 1.

### 4.2. Graph Construction and Preprocessing

Given a syndrome $s$, we construct a complete graph $G = (V, E)$ where $V$ is the set of active defects.

**Node Features:** Each node acts as an index into a feature matrix $A$ of size $N \times 2d_{hidden}$, where $N$ is the total number of stabilizers in the code. For every stabilizer $i$, we define a raw feature vector: $\mathbf{a}_i = [x_i, y_i, \tau_i, \rho_i, \text{PE}_i] \in \mathbb{R}^{d_x}$ where $\mathbf{p}_i \triangleq (x_i, y_i)$ are the 2D lattice coordinates,

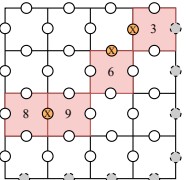 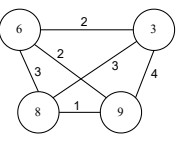

*(a)* 3 physical bit-flips $(X)$ yielding 4 syndrome defects

*(b)* Defects mapped to a complete weighted graph

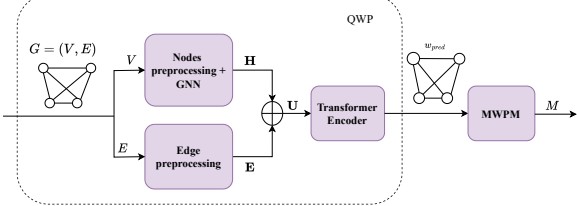

*(c)* The full integration of the syndrome graph, QWP, and MWPM.

*Figure 1.* **Overview of the proposed decoding pipeline.** (a) Three physical errors on the lattice generate four syndrome defects. (b) These defects form the vertices of a complete graph used for matching. (c) The complete NMWPM architecture processes this graph to predict dynamic edge weights for the final correction.

$\tau_i \in \{X, Z\}$ denotes the stabilizer type using a one hot encoded vector, $\rho_i$ represents the Euclidean distance to the lattice center for the Toric Code $(L_{code}/2, L_{code}/2)$, defined as $\sqrt{(x_i - L_{code}/2)^2 + (y_i - L_{code}/2)^2}$. For the Rotated Surface Code, $\rho_i$ is defined as the average coordinate of the physical qubits supported by the stabilizer. Finally, $\text{PE}_i$ is a positional encoding vector (Kipf & Welling, 2017). Crucially, we construct this matrix $A$ for *all* stabilizers, not just the active defects (see Appendix D for details on virtual node augmentation). While inactive stabilizers are assigned zero-vectors for their geometric features $(\mathbf{p}, \tau, \rho)$, they still retain their calculated positional encoding $\text{PE}_i$. Additionally, we initialize a learnable embedding table $\mathbf{R} \in \mathbb{R}^{N \times d_{hidden}}$. For every stabilizer $i$, we retrieve a unique embedding $\mathbf{r}_i = \mathbf{R}[i]$, ensuring the model retains global lattice context. The raw geometric features are processed to a projected subspace $d_{sub} = d_{hidden}/4$. The features $\mathbf{f} \in \{\mathbf{p}_i, \rho_i, \text{PE}_i\}$ are each passed through a two-layer Multi-Layer Perceptron (MLP) with ReLU activation (Nair & Hinton, 2010) to yield $\tilde{\mathbf{f}} \in \mathbb{R}^{d_{sub}}$, while the stabilizer type $\tau_i$ undergoes a single linear projection to obtain $\tilde{\tau}_i \in \mathbb{R}^{d_{sub}}$. The final node representation $\mathbf{a}_i$ is obtained by concatenating (denoted with $\parallel$) the stabilizer embedding $\mathbf{r}_i$ with these processed geometric contexts: $\mathbf{a}_i = [\tilde{\mathbf{p}}_i \parallel \tilde{\tau}_i \parallel \tilde{\rho}_i \parallel \widetilde{\text{PE}}_i \parallel \mathbf{r}_i] \in \mathbb{R}^{2d_{hidden}}$.

**Edge Features and Processing:** For every connected pair of nodes $v_i$ and $v_j$, we consider directed edges in both directions $(v_i \to v_j$ and $v_j \to v_i)$. For the directed edge from

$v_i$ to $v_j$ we first construct a raw feature vector capturing the relative geometry: $\mathbf{e}_{ij} = [d_{ij}, \Delta x_{ij}, \Delta y_{ij}, \tau_{edge}] \in \mathbb{R}^{d_e}$ where $d_{ij}$ is the graph distance, defined as the Manhattan distance between the lattice coordinates of nodes $i$ and $j$, given by $|x_i - x_j| + |y_i - y_j|$. $\Delta x, \Delta y$ are coordinate differences, and $\tau_{edge} \in \{0, 1\}$ is a binary flag indicating the error type associated with the edge. To generate the final edge representation, the discrete graph distance $d_{ij}$ is mapped to a learnable embedding vector $\mathbf{e}_{dist} \in \mathbb{R}^{d_{hidden}}$. Simultaneously, the remaining geometric features are processed by a 2-layer MLP with ReLU activations and Layer Normalization (Ba et al., 2016), projecting them to a vector $\mathbf{e}_{geo} \in \mathbb{R}^{d_{hidden}/2}$. Finally, these two vectors are concatenated to yield the refined edge embedding:

$$\mathbf{e}'_{ij} = [\mathbf{e}_{dist} \parallel \mathbf{e}_{geo}] \in \mathbb{R}^{\frac{3}{2} d_{hidden}} \qquad (7)$$

**GNN Input Preprocessing:** To fully utilize the available signal, we employ a modulated syndrome strategy. We define the modulated syndrome vector $\hat{s} \in \{-1, 1\}^N$ by mapping the binary measurements $\{0, 1\}$ to $\{-1, 1\}$, thereby preserving the structural context of non-defected stabilizers. The node feature matrix is then updated by multiplying each row $i$ (corresponding to stabilizer $i$) by its corresponding scalar modulated syndrome value $\mathbf{a}'_i = \hat{s}_i \cdot \mathbf{a}_i$ yielding an updated feature vector $\mathbf{a}'_i \in \mathbb{R}^{2d_{hidden}}$ that encodes both the stabilizer's geometric identity and its activation state. Finally, to manage computational complexity, the features are projected down to half their dimensionality before entering the GNN backbone. We apply a linear transformation followed by Layer Normalization to map the feature vectors from $\mathbb{R}^{2d_{hidden}} \to \mathbb{R}^{d_{hidden}}$. The resulting vector serves as the initial input to the GNN, denoted as $\mathbf{h}_i^{(0)}$. This preprocessing sequence is depicted in Fig. 1c.

### 4.3. Quantum Weight Predictor

#### 4.3.1. LOCAL PROCESSING: GNN BLOCK

The first stage of our neural backbone processes the local topology of the syndrome graph using a GNN. We employ a stack of $L_{layers}$ identical layers based on Graph Transformer operator (Shi et al., 2021). We adopt a Pre-Layer Normalization architecture, which has been shown to improve training stability in Transformers (Xiong et al., 2020). Let $\mathbf{h}_i^{(l)} \in \mathbb{R}^{d_{hidden}}$ denote the feature vector of node $i$ at layer $l$. The processing flow for a single layer consists of a Multi-Head Self-Attention (MHSA) mechanism followed by a Feed-Forward Network (FFN). First, the inputs are normalized and processed by the Graph Transformer operator. To maintain the feature dimension $d_{hidden}$ across layers, we utilize an ensemble of $K$ attention heads and average their outputs. We denote the neighborhood of node $i$ as $\mathcal{N}(i)$. The normalized features are given by $\hat{\mathbf{h}}_i = \text{LayerNorm}(\mathbf{h}_i^{(l)})$. For each head $k \in \{1, \ldots, K\}$,

we first compute the attention coefficients $\alpha_{ij}^{(k)}$ via scaled dot-product attention. We then aggregate the neighborhood information by averaging the weighted messages from all $K$ heads to produce a unified context vector:

$$\mathbf{m}_i = \frac{1}{K} \sum_{k=1}^{K} \sum_{j \in \mathcal{N}(i)} \alpha_{ij}^{(k)} (\mathbf{W}_2^{(k)} \hat{\mathbf{h}}_j + \mathbf{b}_2^{(k)}) \qquad (8)$$

To dynamically regulate the information flow, we compute a gating coefficient $\beta_i$. Let $\tilde{\mathbf{h}}_i = \mathbf{W}_1 \hat{\mathbf{h}}_i + \mathbf{b}_1$ denote the projected self-features. The gate considers the context vector, the self-features, and their difference:

$$\beta_i = \text{sigmoid} \left( \mathbf{w}_3^\top \left[ \mathbf{m}_i \parallel \tilde{\mathbf{h}}_i \parallel (\mathbf{m}_i - \tilde{\mathbf{h}}_i) \right] \right) \qquad (9)$$

The final updated node embedding $\mathbf{z}_i$ is obtained by gating the self-features against the neighborhood message:

$$\mathbf{z}_i = \beta_i \tilde{\mathbf{h}}_i + (1 - \beta_i) \mathbf{m}_i \qquad (10)$$

The matrix $\mathbf{W}_2^{(k)} \in \mathbb{R}^{d_{hidden} \times d_{hidden}}$ and bias vector $\mathbf{b}_2^{(k)} \in \mathbb{R}^{d_{hidden}}$ are head-specific parameters. $\mathbf{W}_1 \in \mathbb{R}^{d_{hidden} \times d_{hidden}}$ and $\mathbf{b}_1 \in \mathbb{R}^{d_{hidden}}$ are projection weights, while $\mathbf{w}_3 \in \mathbb{R}^{3d_{hidden}}$ is the gating weight. The final output is added to the input residual:

$$\mathbf{h}'_i = \mathbf{z}_i + \mathbf{h}_i^{(l)} \qquad (11)$$

Subsequently, the updated node features pass through a FFN. Consistent with the pre-layer normalization architecture, the input is first normalized before passing through a two-layer MLP with a Gaussian Error Linear Unit (GELU) (Hendrycks & Gimpel, 2016) activation. This network projects the hidden dimension $d_{hidden}$ to an intermediate dimension of $4d_{hidden}$ before projecting it back to $d_{hidden}$. The output $\mathbf{h}_i^{(l+1)} \in \mathbb{R}^{d_{hidden}}$ is computed via a residual connection:

$$\mathbf{h}_i^{(l+1)} = \text{FFN}(\text{LayerNorm}(\mathbf{h}'_i)) + \mathbf{h}'_i \qquad (12)$$

#### 4.3.2. GLOBAL PROCESSING: TRANSFORMER ENCODER

To predict the probability of an edge being part of the error chain, we combine information from the two incident nodes and the edge itself. We construct a composite representation $\mathbf{u}_{ij}$ for every edge $(i, j)$ by concatenating the processed node embeddings and the processed edge embedding:

$$\mathbf{u}_{ij} = [\mathbf{h}_i^{(l+1)} \parallel \mathbf{h}_j^{(l+1)} \parallel \mathbf{e}'_{ij}] \in \mathbb{R}^{2d_{hidden} + \frac{3}{2} d_{hidden}} \qquad (13)$$

This vector $\mathbf{u}_{ij}$ is normalized and fed into a standard Transformer Encoder, denoted as $\mathcal{T}_\theta(\cdot)$. The self-attention mechanism allows the model to weigh the importance of different edges against each other globally, effectively reasoning

about competing error chains across the lattice, as presented in Figure 1c. The output is:

$$\mathbf{o}_{ij} = \mathcal{T}_\theta(\text{LayerNorm}(\mathbf{u}_{ij})) \tag{14}$$

The output is then passed through a final projection layer parameterized by $\mathbf{w}_{out} \in \mathbb{R}^{2d+\frac{3}{2}d_{hidden}}$ and bias $b \in \mathbb{R}$, followed by a Sigmoid activation:

$$p_{ij} = \sigma(\mathbf{w}_{out}^\top \mathbf{o}_{ij} + b) \tag{15}$$

yielding a probability $p_{ij} \in [0, 1]$ for every edge in the fully connected graph. These probabilities are subsequently converted into weights to guide the classical matching process. The complete inference pipeline, incorporating the neural network predictions with the MWPM decoder, is formalized in Algorithm 1 and Figure 1c .

### 4.3.3. DECODING:

The model outputs probability scores for directed edges. However, the standard MWPM algorithm operates on an undirected graph. To accommodate this, we aggregate the predictions for the two directions of each edge by taking the maximum probability:

$$p'_{ij} = \max(p_{ij}, p_{ji}) \tag{16}$$

We then convert these unified probabilities into weights suitable for the MWPM algorithm:

$$w_{ij} = -\ln(p'_{ij}) \tag{17}$$

Edges with high probability ($p' \approx 1$) result in weights close to 0, making them highly attractive to the minimization algorithm. Conversely, the logarithmic transformation imposes a steep penalty on low-probability edges ($p' \approx 0$), assigning them large positive weights that effectively discourage their selection during matching. During inference, the predicted weights $w_{ij}$ are passed to the MWPM algorithm. The resulting matching determines the correction operator applied to the quantum state.

### 4.4. Training

**Loss Function:** We train the network using a composite loss function designed to maximize accuracy while enforcing prediction confidence. Let us denote the number of edges in the decoding graph $d_e$. The primary component of the loss function is the BCE between the predicted probabilities $\mathbf{p} \in \mathbb{R}^{2d_e}$ and the ground truth error edges $\mathbf{y} \in \mathbb{R}^{2d_e}$. To mitigate uncertainty and push the model towards decisive predictions, we add an entropy regularization term, formulated as the BCE of the probability vector with itself:

$$\mathcal{L} = \text{BCE}(\mathbf{p}, \mathbf{y}) + \lambda \cdot \text{H}(\mathbf{p}) \tag{18}$$

where $H$ denotes entropy and $\lambda$ is a hyperparameter governing the regularization strength. This term is particularly critical for the downstream MWPM decoder. Because the algorithm minimizes the total additive weight of the matching, enforcing a sharp dichotomy in predicted probabilities pushes weights towards zero or infinity, thereby preventing the aggregation of small uncertainties that would otherwise obscure the optimal error.

---

**Algorithm 1** Neural MWPM

**Require:** Syndrome $S$
**Ensure:** The output $M$ is a set of edges such that every vertex in $S$ is incident to exactly one edge in $M$.
1: $\mathcal{D} \leftarrow \text{QWP}(S)$
2: Initialize weights set $W \leftarrow \emptyset$
3: **for** each probability $p_{ij} \in \mathcal{D}$ **do**
4:     $w_{ij} \leftarrow -\log(p_{ij})$
5:     $W \leftarrow W \cup \{w_{ij}\}$
6: **end for**
7: $M \leftarrow \text{MWPM}(S, W)$
8: **return** Matching $M$

---

**Algorithm 2** Ground Truth Construction

**Require:** Error configuration $e$
**Ensure:** Ground truth matching $M$
1: $\mathcal{C} \leftarrow \text{ClusterErrors}(e)$
2: $M \leftarrow \emptyset$
3: **for** each cluster $C \in \mathcal{C}$ **do**
4:     $S_{local} \leftarrow \text{GetEndpoints}(C)$
5:     $M_{local} \leftarrow \text{MWPM}(S_{local}, \text{weight} = \text{distance})$
6:     $M \leftarrow M \cup M_{local}$
7: **end for**
8: **if** LogicalError$(e, M)$ **then**
9:     $M \leftarrow \text{FindValidPermutation}(\mathcal{C}, M)$
10: **end if**
11: **return** $M$

---

**Ground Truth Generation:** To train the network, we require a set of binary labels $\mathbf{y}$ where $y_{ij} = 1$ if an edge $(i, j)$ belongs to the optimal error correction chain, and 0 otherwise. Generating these labels is non-trivial due to the degeneracy of topological codes. We employ a heuristic clustering algorithm to approximate the true error chain used in the simulation. We decompose the global error configuration into independent clusters by grouping qubits connected via shared stabilizers. From these clusters, we filter out stabilizers that interact with an even number of errored qubits (even parity), retaining only the endpoints that correspond to active syndrome defects. Isolated pairs of defects are treated as direct matches. For larger, more complex clusters, we employ a localized MWPM with Manhattan distance weights. If this solution results in a logical error, we

iteratively permute the matching assignments within each cluster until a valid correction is obtained. In the context of the Rotated Surface code, boundary handling necessitates the introduction of a virtual node whenever a cluster contains an odd number of endpoints. In instances where the heuristic method fails, we resort to a timed brute-force search, ensuring a valid ground truth is retrieved without incurring prohibitive computational costs for outliers. Further details regarding timeout protocols, data filtering, and label purity are provided in Appendix C.

## 5. Experiments and Results

### 5.1. Experimental Setup

To validate the efficacy of NMWPM, we focus our analysis on the following codes: the periodic Toric (Kitaev, 1997) and the Rotated Surface (Bombín & Martin-Delgado, 2007). Detailed descriptions of the code construction and stabilizer geometry are provided in Appendix A. We assess the robustness of the decoder under two canonical error channels: the standard independent noise and the more challenging depolarizing noise. We benchmark our performance against four key baselines. First, the standard MWPM algorithm (Fowler, 2015), which serves as the gold-standard classical baseline. Second, to contextualize our results within the machine learning landscape, we evaluate our method against QECCT (Choukroun & Wolf, 2024) and Belief Propagation with Order-2 Ordered Statistics Decoder (BPOSD-2) (Roffe et al., 2020). Finally, we compare against a correlated variant of MWPM (MWPM-Corr) (Fowler, 2013). For the neural architecture, we configured the model with a hidden dimension of $d_{hidden} = 128$. The GNN backbone is composed of $L_{layers} = 4$ layers, with $K = 4$ attention heads. For the Transformer Encoder, we used $L_{enc} = 2$ layers. The final MWPM step was executed using PyMatching (Higgott & Gidney, 2025). We trained the model using the Adam optimizer (Kingma & Ba, 2015) with a batch size of 32 and an initial learning rate of $9 \times 10^{-5}$, with cosine annealing decay reducing it to $1 \times 10^{-5}$. Each epoch processes 500 mini-batches. The loss function incorporates an entropy regularization term weighted by the hyperparameter $\lambda = 0.01$. This configuration was maintained identically across all evaluated code sizes.

### 5.2. Results and Evaluation Metrics

We assess decoder performance via three metrics, LER, algorithmic complexity, and the threshold ($p_{th}$), the physical error rate below which increasing code distance ($L_{code}$) reduces logical errors.

We evaluate the performance of the proposed decoder on the Toric code for $L_{code} \in \{6, 8, 10, 12\}$ and on the Rotated Surface Code for $L_{code} \in \{5, 7, 9\}$. QECCT eval-

uations for $L_{code} = 12$ are omitted due to GPU memory constraints imposed by the quadratic growth of its attention matrix. First, we analyze the Toric code under the depolarizing noise model. As illustrated in Figure 2a-d, our NMWPM decoder demonstrates a substantial reduction in LER compared against both the standard MWPM baseline and BPOSD-2. This advantage is particularly pronounced at $L_{code} = 10$, where we achieve a $17 - 50\%$ reduction in LER for physical error rates $p > 0.12$. When compared to the state-of-the-art QECCT baseline, our model exhibits superior scaling characteristics: while we observe modest improvements at smaller lattice sizes ($L_{code} = 6, 8$), the performance gap widens significantly at $L_{code} = 10$ in favor of our hybrid approach. This improvement is driven by the model's dual-stage processing: the GNN layers effectively extract local topological features from the syndrome graph, while the Transformer block resolves the global correlations that emerge in larger lattices. This combination allows the decoder to maintain high precision even as the complexity of the error chains increases. Regarding threshold, as shown in Figure 2i we identify a threshold of $17.9\%$ while the maximum likelihood bound is $18.9\%$ (Bombin et al., 2012), outperforming MWPM and BPOSD-2 ($16.0\%$) and QECCT ($17.8\%$). Under the independent noise model for the Toric code, our method maintains a consistent, yet smaller, advantage over the baselines. In terms of threshold characteristics, as illustrated in Figure 2h we recover a value of $10.95\%$ for independent noise, aligning closely with the theoretical maximum likelihood bound of $11.0\%$, outperforming MWPM ($10.3\%$) (Wang et al., 2003; Higgott, 2022), BPOSD-2 ($10.8\%$) and QECCT ($10.7\%$) with our implementation. Following the Toric code analysis, we evaluate the rotated surface code under depolarizing noise (Figure 2e-g). NMWPM consistently outperforms classical baselines and yields marginal gains over QECCT, confirming that our hybrid architecture generalizes to rotated geometries. In terms of threshold performance, NMWPM maintains its robust capabilities on the rotated lattice, reaching a threshold of $17.7\%$. Surpassing QECCT ($17.2\%$) and classical baselines, BPOSD-2 ($14.1\%$ based on our experiments) and MWPM ($14.0\%$) (deMarti iOlius et al., 2024). Finally, to validate our approach under more realistic conditions, we evaluated the framework on the repetition code under circuit-level noise using Stim (Gidney, 2021), where the number of measurement rounds equals the code distance ($L_{code}$ = rounds). As shown in Figure 3, NMWPM consistently outperforms the standard MWPM. The reduction in LER becomes increasingly prominent at higher error rates. Appendix F provides a detailed comparison of our depolarizing noise threshold against both neural and classical baselines.

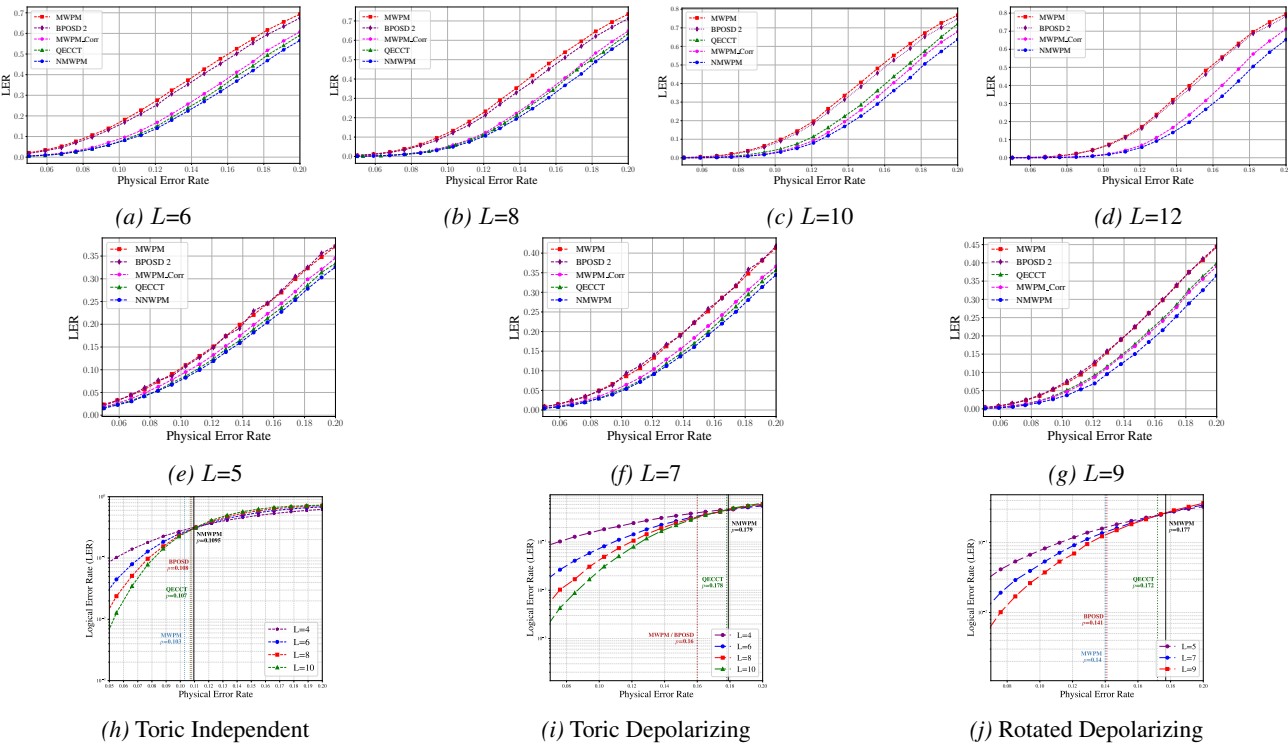

*Figure 2.* Comprehensive Error Analysis. **(a)-(d)** LER vs. Physical Error Rate for Toric Code. **(e)-(g)** LER vs. Physical Error Rate for Rotated Surface Code. **(h)-(j)** Error Threshold Analysis across different noise models and geometries.

## 5.3. Computational Complexity

QWP complexity scales from sparse graph-based feature extraction to dense global attention. The process begins with node feature projection scaling as $O(N d_{\text{hidden}}^2)$, where $N$ is the total number of stabilizer nodes. While the $L_{\text{layers}}$ layers of the GNN block maintain an efficient complexity of $O(L_{\text{layers}} \cdot (N d_{\text{hidden}}^2 + E d_{\text{hidden}}))$ by leveraging the sparse connectivity of the graph, the subsequent Transformer Encoder treats all $E$ edges in the defect graph as a sequence of tokens for dense self-attention. This operation introduces a quadratic dependency on the number of edges, resulting in an overall theoretical complexity of $O(E^2 d_{\text{hidden}} + E d_{\text{hidden}}^2)$. Despite this scaling, the design choice leverages the global context aggregation power of the self-attention mechanism, offering the superior representational capacity essential for accurately resolving complex, non-local error patterns within the defect graph.

## 5.4. Parameter Efficiency

While classical decoders like MWPM and BPOSD-2 are non-parametric, neural-augmented decoders improve decoding by learning from the specific error distributions. A critical advantage of the proposed framework is its architectural scalability. The QECCT baseline exhibits rapid inflation in model size as the code distance increases, reaching 6.71M

parameters at $L_{code} = 10$ for depolarizing noise. In stark contrast, our NMWPM maintains a nearly constant footprint of approximately 3.9M parameters across all tested lattice sizes. This efficiency ensures model viability for larger code distances without prohibitive memory requirements.

| Code Distance | NMWPM (M) | QECCT (M) |
|---|---|---|
| $L_{code} = 6$ | 3.99 | 1.90 |
| $L_{code} = 8$ | 3.99 | 3.42 |
| $L_{code} = 10$ | 4.00 | 6.64 |

*Table 1.* Comparison of parameter counts in millions between NMWPM and QECCT across different code distances under depolarizing noise.

## 6. Model Analysis

### 6.1. Weight Distribution Dynamics

As illustrated in Figure 4a, the model's predictive confidence undergoes a significant transformation throughout the training process. Initially, the probability distribution is broad, reflecting high uncertainty and a lack of clear internal representation of noise correlations, with many edges assigned mid-range probabilities. Upon convergence, the distribution exhibits a strong polarization; the vast majority of edges are

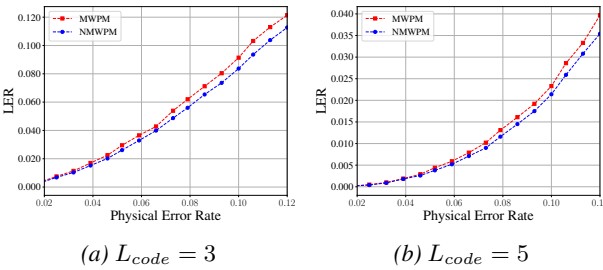

*(a) $L_{code} = 3$*          *(b) $L_{code} = 5$*

*Figure 3.* Repetition Code under Circuit-Level Noise. LER vs. Physical Error Rate.

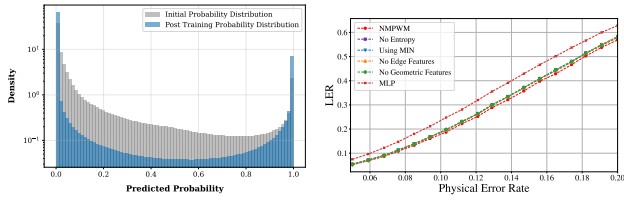

*(a)* Weight distribution evolution     *(b)* Ablation Study (LER)

*Figure 4.* **Analysis Results.** (a) weights evolving to a bimodal distribution, showing higher confidence ($L = 7$) (b) Ablation results showing NMWPM outperforming alternative architectures.

pushed toward a probability of zero, while likely error edges concentrate near one. This polarization is a key indicator of the model's success. By effectively filtering the matching graph through these high-contrast weights, NMWPM reduces the search space for the classical MWPM algorithm, allowing it to resolve complex error patterns with higher accuracy than standard baselines.

### 6.2. Generalization and Transfer Learning

To demonstrate that NMWPM learns robust decoding priors rather than overfitting to specific noise realizations, we evaluated its generalization capabilities across multiple dimensions. The model generalizes effectively across varying code topologies (toric, rotated surface, and repetition codes) and diverse noise models (independent, depolarizing, and circuit-level). Furthermore, NMWPM is highly robust to variations in the physical error rate $p$. Beyond interpolating within the training distribution, it reliably extrapolates to unseen error rates strictly outside the training regime (evaluated at $p = 0.03$ and $p = 0.23$), consistently maintaining lower LER than MWPM. Finally, we demonstrate the transferability of the learned representations. By initializing an $L_{code} = 8$ model using the Transformer weights from a pre-trained $L_{code} = 10$ model, alongside randomly initialized GNN weights, the model rapidly adapts to the new spatial dimensions. Fine-tuning this transferred model for only 20 epochs yields results that significantly outperform the standard MWPM, indicating that the Transformer learns invariant features for edge weight prediction. Detailed quantitative analysis is provided in Appendix E.

### 6.3. Ablation Study

We validated our architecture through an ablation study on the toric code ($L_{code} = 4$) under depolarizing noise (Figure 4b). Removing either geometric embeddings or edge features consistently increased LER, confirming their importance for decoding. Similarly, omitting entropy regularization ("No Entropy") caused a slight degradation in performance suggests that regularization is leading to more robust matching. We also verified our aggregation strategy.

Replacing our maximum aggregation $p'_{ij}$ with a conservative minimum ("Using MIN") reduced performance, suggesting that the latter overly suppresses asymmetric error signals. Finally, substituting the Transformer backbone with an MLP caused the most significant drop in accuracy, justifying our architectural complexity. This confirms that self-attention is essential for capturing long-range dependencies between syndrome defects, which a simple MLP fails to model.

## 7. Conclusion

We introduced our NMWPM framework, which formulates the decoding problem as a differentiable edge-weight prediction task. Our approach demonstrates superior scalability, significantly outperforming standard baselines under well studied noise regimes across two topological codes. Ultimately, these results highlight the efficacy of augmenting classical decoders with learned priors, encouraging the broader adoption of hybrid methodologies in QEC.

## Acknowledgements

This work was supported by the D.E. Koshland Jr. Family Career Development Chair in Advanced Technologies in Electrical and Computer Engineering

## Impact Statement

This research explores the intersection of machine learning and quantum information science, focusing on enhancing the reliability and speed of error decoding. While our work contributes to the realization of fault-tolerant quantum computing, a field with transformative potential for medicine and material science, we acknowledge its dual-use nature. Specifically, the advancement of quantum hardware poses a known challenge to current cryptographic standards. However, the development of robust error correction is a fundamental prerequisite for any practical quantum application. We believe that advancing decoding capabilities alongside the global transition toward post-quantum cryptography ensures that the benefits of quantum computing can be realized while mitigating its security risks.

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

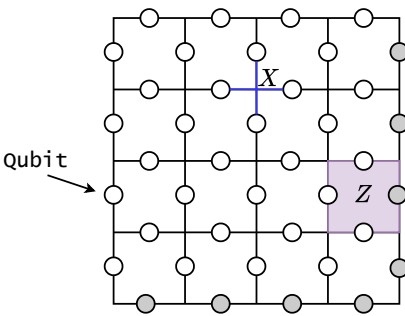

*Figure 5.* **Schematic of Code Topology:** Layout of the Toric code ($L = 4$). The gray qubits denote the periodic connections required for the torus geometry, and examples of the distinct stabilizer generators are marked.

## A. Surface Codes

In this section, we detail a prominent surface code architecture, selected due to its widespread popularity, the Toric code (Kitaev, 1997).

The Toric code encodes $k = 2$ logical qubits using $n = 2L_{code}^2$ physical qubits positioned on the edges of a lattice with periodic boundary conditions. Its stabilizer generators are partitioned into two geometrically distinct groups: vertex stabilizers, formed by the product of Pauli-$X$ operators on the four edges adjacent to a vertex; and plaquette stabilizers, formed by the product of Pauli-$Z$ operators on the four edges bounding a lattice face. This structure yields a total of $m = 2L_{code}^2 - 2$ generators, comprising $L_{code}^2 - 1$ vertex stabilizers and $L_{code}^2 - 1$ plaquette stabilizers.

We evaluate this architecture under two standard noise models. First, the independent noise model assumes uncorrelated bit-flip ($X$) and phase-flip ($Z$) errors with equal probability, allowing $X$ and $Z$ syndromes to be decoded separately. Second, the depolarizing noise model accounts for correlations by assigning equal probability $p/3$ to each non-identity Pauli operator, such that $\Pr(X) = \Pr(Z) = \Pr(Y) = p/3$, $Y = iXZ$.

## B. Model and Training Details

Our training methodology randomly samples noise within the physical error rate testing range to ensure robust generalization across different noise regimes. The hybrid model architecture employs 4 TransformerConv (Shi et al., 2021) layers followed by 2 Transformer Encoder layers, with a shared embedding dimension of $d_{hidden} = 128$ and $K = 4$ attention heads used in both the GNN and encoder blocks. The loss function incorporates an entropy regularization term weighted by the hyperparameter $\lambda = 0.01$ to encour-

age prediction confidence.

We optimize the model using the Adam optimizer (Kingma & Ba, 2015) with a batch size of 32. Training spans 200–1000 epochs, where each epoch processes 500 mini-batches. The learning rate is initialized at $9 \times 10^{-5}$, with cosine annealing decay reducing it to a minimum of $1 \times 10^{-5}$ by the end of training. All experiments were conducted on a 48GB NVIDIA L40 GPU. We utilize the toric code implementation from Krastanov & Jiang (2017) (Krastanov & Jiang, 2017). Figure 7 provides overview of the network architecture. This visual representation supplements the methodology section.

## C. Ground Truth Heuristic

The procedure for generating ground truth matchings, as outlined in Section 4, relies on a heuristic clustering approach followed by a localized fallback search. Below, we detail the computational considerations, data filtering protocols, and the handling of code degeneracy during this process.

**Search Efficiency and Timeout Protocol:** The proposed heuristic efficiently resolves the vast majority of error configurations. For example, when evaluating the Toric code under the depolarizing noise model ($L_{code} = 8$), only $0.6\%$ of the dataset necessitates the brute-force search fallback. To balance dataset generation time with data retention, we impose a strict 10-second timeout per sample during this phase.

**Data Filtering and Label Purity:** Samples that exceed the 10-second computational limit, amounting to approximately $0.2\%$ of the generated data, are discarded. By exclusively training on explicitly verified matchings, we effectively eliminate residual label noise from the training process. We emphasize that this filtering procedure is strictly confined to the training data generation phase, *absolutely no samples are discarded during inference or evaluation*, ensuring a rigorous and unbiased benchmark against classical decoders.

**Degeneracy and Non-Trivial Labels:** Topological codes exhibit high degeneracy, meaning multiple logically equivalent matchings exist for a given syndrome. Our deterministic heuristic assigns the first valid matching it encounters as the definitive ground truth. This early-stopping approach provides a clean, consistent training signal while minimizing computational overhead. Crucially, because these heuristically derived labels can deviate from the standard shortest-path solutions generated by classical MWPM, they naturally yield non-trivial solutions. By training on these labels, the neural network learns to identify and correct complex error chains where standard MWPM typically fails.

## D. Rotated Surface Code Boundary Treatment

To ensure complete reproducibility, we clarify our treatment of boundary conditions for the Rotated Surface code. Unlike the periodic Toric code, the Rotated Surface code features open boundaries where error chains can terminate. To accommodate this within our framework, a virtual boundary node is explicitly incorporated into the learned syndrome graph. Specifically, we augment the node feature matrix with an additional row representing this virtual node. To maintain a complete graph topology, this virtual node is connected to all active defect nodes. This structural modification enables the model to effectively capture spatial correlations between both bulk and boundary errors. Consequently, the QWP predicts the probability of an error chain connecting each active defect directly to the boundary, assigning a learned weight to every defect-to-boundary edge prior to the matching phase.

## E. Detailed Analysis of Interpolation and Extrapolation

A critical property of our decoder is its robustness to unseen noise distributions. NMWPM was trained on physical error rates in the $[0.05, 0.20]$ range. Here, we provide an expanded analysis of the interpolation and extrapolation capabilities of NMWPM.

**Interpolation:** The model was trained on 9 distinct physical error rates. It was subsequently evaluated on 18 points within that training range, covering a much denser set of noise probabilities. The model demonstrated excellent interpolation performance, generalizing to unseen intermediate noise probabilities, as seen in Figure 2.

**Extrapolation:** To assess extrapolation capability, performance on error rates strictly outside the training distribution, we evaluated the model on the Toric ($L_{code} = 8$) and Rotated ($L_{code} = 7$) codes. The model was evaluated at unseen error rates $p = 0.03$ and $p = 0.23$. As detailed in Tables 3 and 4, NMWPM maintained high performance in both regimes. This consistently lower LER confirms that the learned priors generalize effectively.

### E.1. Transfer Learning Capabilities

To explicitly evaluate the transferability of the learned representations, we conducted a transfer learning experiment across different code distances. Specifically, we initialized an NMWPM model for a smaller code size ($L_{code} = 8$) using the pre-trained Transformer weights derived from a larger model trained on $L_{code} = 10$. The GNN weights for the $L_{code} = 8$ model were initialized randomly. As detailed in Table 2, fine-tuning this transferred model for only 20 epochs yields a logical error rate that significantly

outperforms the MWPM baseline. This rapid convergence indicates that the Transformer effectively learns generalizable features for syndrome graph edge weight prediction that are not strictly bound to the spatial dimensions of the original training code.

| p | MWPM | Transferred NMWPM |
|---|---|---|
| 0.103 | 0.1333 | 0.0602 |
| 0.129 | 0.2906 | 0.1654 |
| 0.191 | 0.6940 | 0.5752 |

*Table 2.* LER for the transfer learning experiment. The model was transferred from an $L_{code} = 10$ pre-trained model to $L_{code} = 8$ and fine-tuned for only 20 epochs.

| p | MWPM | NMWPM (Ours) |
|---|---|---|
| 0.030 | 0.0005 | 0.0001 |
| 0.230 | 0.8324 | 0.7568 |

*Table 3.* LER extrapolation results for the Toric code ($L_{code} = 8$) under Depolarizing noise.

| p | MWPM | NMWPM (Ours) |
|---|---|---|
| 0.030 | 0.0015 | 0.0007 |
| 0.230 | 0.5048 | 0.4475 |

*Table 4.* LER extrapolation results for the Rotated Surface code ($L_{code} = 7$) under Depolarizing noise

## F. Comparative Threshold Analysis

Table 5 presents a comparison of depolarizing noise thresholds against additional neural and classical baselines. As shown, NMWPM achieves the highest threshold.
**Note:** Unless otherwise specified, all reported threshold percentages reflect performance Toric or Rotated Surface codes evaluated under the standard depolarizing noise model.

| Decoder / Paradigm | Reported Threshold ($\downarrow$) |
|---|---|
| NMWPM (Toric Code) | **17.9%** |
| NMWPM (Rotated Code) | **17.7%** |
| QECCT (Choukroun & Wolf, 2024) | 17.8% |
| Astra (2024) (Maan & Paler, 2025) | ~17.0% |
| Recursive MWPM (iOlius et al., 2023) | ~16.5% |
| SU-NetQD (2025) (Zhang et al., 2025) | 16.3% |
| ML+UF (2022) (Meinerz et al., 2022) | 16.2% |
| MWPM (Classical Baseline) (Wang et al., 2010) | 16.0% |
| BP-OSD (Classical Baseline) (Roffe et al., 2020) | ~16.0% |
| UIUF (2024) (Lin & Lai, 2025) | 15.6% |

*Table 5.* Comparative Thresholds for Surface Codes (Depolarizing Noise)

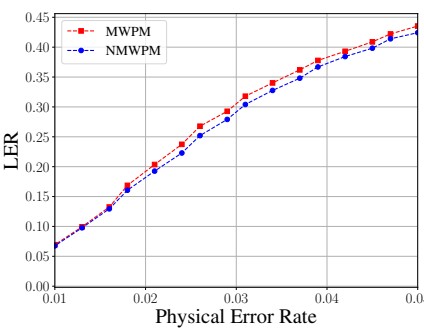

Figure 6. LER vs. Physical Error Rate for Rotated X Memory

## G. Inference Time Benchmark

To evaluate the computational efficiency of our approach, we conducted an inference-time benchmark using a single NVIDIA L40 GPU on the Toric code under depolarizing noise. Table 6 presents a comparison of the inference times between our proposed NMWPM and the QECCT baseline for code distances $L_{code} = 6$ and $L_{code} = 10$.

| $L$ | QECCT Total | NMWPM GPU | NMWPM CPU | NMWPM Total |
|---|---|---|---|---|
| 6 | 7.00 ms | 2.23 ms | 0.24 ms | **2.47 ms** |
| 10 | 20.1 ms | 2.82 ms | 1.19 ms | **4.01 ms** |

Table 6. Inference time comparison between QECCT and NMWPM. The NMWPM runtime is broken down into GPU and CPU processing times. All measurements were conducted on a single NVIDIA L40 GPU.

As shown in Table 6, NMWPM consistently achieves faster total inference times than QECCT, and this advantage becomes more pronounced as the code distance increases. At $L_{code} = 6$, NMWPM's total runtime is 2.47 ms compared to 7.00 ms for QECCT. When scaling to $L_{code} = 10$, QECCT's runtime increases significantly to 20.1 ms. In contrast, NMWPM scales much more efficiently, with the total runtime increasing to only 4.01 ms. This efficiency is achieved because the majority of the computational load is processed on the GPU, which scales well, increasing from 2.23 ms to only 2.82 ms between $L_{code} = 6$ and $L_{code} = 10$.

## H. Evaluation on Rotated X Memory Circuit Noise

We present an additional evaluation of the NMWPM decoder under circuit-level noise. We simulated a rotated surface code for an X memory experiment with a code distance of $L_{code} = 3$ and 3 syndrome measurement rounds. Figure 6 shows the LER of our model compared to a standard MWPM baseline across physical error rates ranging from $p = 0.01$ to $p = 0.05$.

In this specific setting, NMWPM achieves a lower or equal

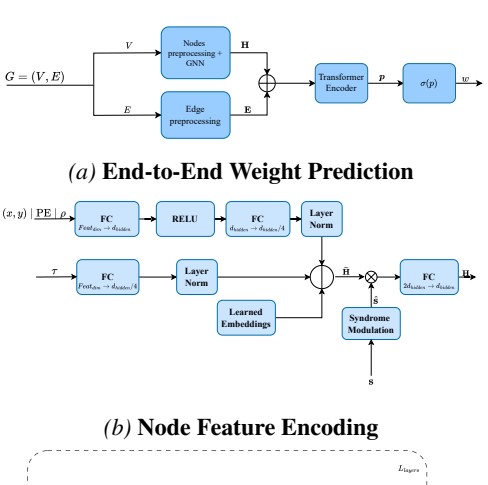

*(a)* **End-to-End Weight Prediction**

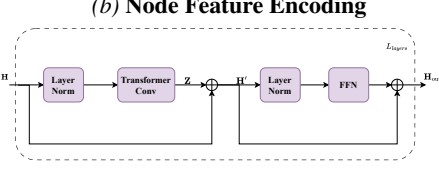

*(b)* **Node Feature Encoding**

*(c)* **Graph Transformer Layer**

Figure 7. **Architectural Schematic.** (a) The full inference pipeline. (b) Preprocessing of syndrome data. (c) GNN block.

LER than the baseline. While both models perform similarly at the lower end of the error regime ($p = 0.01$), the performance gap widens as the error rate increases.

