# OpenReview forum: "Neural Minimum Weight Perfect Matching for Quantum Error Codes"
_ICML.cc/2026/Conference — ICML 2026 regular_

### Official Review · Reviewer_E3CG · 2026-03-12

**Soundness:** 2
**Presentation:** 1
**Significance:** 2
**Originality:** 2
**Overall Recommendation:** 3
**Confidence:** 4

**Summary:**

This paper proposes Neural Minimum Weight Perfect Matching (NMWPM), a hybrid decoder for quantum error correction that enhances the standard MWPM framework with learned, syndrome-dependent edge weights. The model combines graph neural networks to extract local structure from syndromes with Transformers to capture longer-range dependencies, and uses the resulting representations to predict dynamic edge weights for matching.

**Compliance With Llm Reviewing Policy:**

Affirmed.

**Final Justification:**

I still have concerns about the computational cost of data generation, model training, inference, generalization capability, and theoretical guarantees as the system size grows, as these are crucial for evaluating whether the approach remains viable at larger scales. The current theoretical and numerical evidence is not sufficiently convincing to me.

**Key Questions For Authors:**

- How well does the learned decoder generalize beyond the original training distribution, such as to different noise rates, code distances, or related noise models?
- What are the costs of data generation, training, and inference as the problem scale increases, and is there evidence that the method remains effective and practical at larger scales?

**Limitations:**

- The paper would benefit from a more self-contained introduction to the method. The current explanation should more clearly define the decoding setting, the architecture, the predicted quantities, and the training objective.
- The algorithmic pipeline should be described more explicitly, ideally in a way that makes the interaction between neural prediction and MWPM easy to follow.
- If the model requires substantial retraining or large data-generation cost for each new regime, its practical value may be more limited than the headline threshold numbers suggest.

**Strengths And Weaknesses:**

Strengths
- The paper focused on a meaningful and practical problem in quantum error correction.
- The hybrid design is well motivated at a high level.
- Using GNNs for local syndrome structure and Transformers for global dependencies is an interesting architectural choice.

Weaknesses
- The method needs clearer definition and explanation, especially regarding the neural network components, the logic, and how the overall algorithm operates. At present, these core pieces do not seem sufficiently transparent.
- The paper should provide more evidence about scalability. In particular, the computational cost of data generation, model training, and inference as system size grows is crucial for evaluating whether the approach remains viable at larger scales.
- Need more intuition for the performance improvements would make the paper easier to evaluate and more compelling.

---

> ### Author Rebuttal · Authors · 2026-03-31
>
> We thank the reviewer for their constructive feedback, and address the raised points below.
> Please note that all new figures are available at the following anonymous link:
> https://anonymous.4open.science/r/ICML2026_Rebuttal_1234-1B20sashdfkk/
> ## Weakness 1 - Method Clarity
> We thank the reviewer and provide a structured overview to clarify our methodology:
> 1. **Graph Construction & Preprocessing:** The syndrome forms a complete graph between active defects using a complete stabilizer node matrix. All nodes receive positional encodings (active nodes use preprocessed geometric features, inactive use zeros). Crucially, each matrix row is multiplied by the modulated syndrome ($\pm 1$) (Appendix B, Fig. 5b). Concurrently, edge features are processed via MLP (geometric data) and distance embeddings.
> 2. **Global Feature Extraction:** A Graph Transformer processes the node matrix to capture global syndrome correlations (App. B, Fig. 5c).
> 3. **Prediction & Decoding:** Processed source, target, and edge features are concatenated and passed through a Transformer encoder predicting directed probabilities (sigmoid). For undirected edges: $\max(p_{ij}, p_{ji})$. Weights $w = -\log(p)$ are passed to the classical MWPM decoder.
>
> We would be happy to explain or clarify any specific components that remain unclear.
> ## Weakness 3 - Performance Intuition
> We thank the reviewer for the suggestion to provide more intuition. NMWPM achieves state-of-the-art thresholds by capturing noise-specific correlations, approaching Maximum Likelihood (ML) limits while maintaining the structural guarantees of MWPM. Notably, NMWPM achieves a 17.7% threshold on the rotated surface code under depolarizing noise. In the Toric code, it achieves 10.95% under independent noise (nearly reaching the 11.0% ML limit) and 17.9% under depolarizing noise (ML limit: 18.9%). The table below contains reported depolarizing thresholds on surface codes from strong baselines:
>
> | **Decoder** | **Threshold Noise ($p_{th}$)** |
> | :--- | :--- |
> | NMWPM | 17.9% |
> | QECCT [1] | 17.8% |
> | Astra (2024) [2] | 17.0% |
> | SU-NetQD (2025) [3] | 16.3% |
> | ML+UF (2022) [4] | 16.2% |
> | BP-OSD2 [5] | 16.0% |
> | MWPM [6] | 16.0% |
>
> NMWPM also outperforms other MWPM variants like Recursive MWPM [7], achieving a 17.7% versus 16.5% threshold on the rotated surface code. By combining a Graph Transformer for global features with a Transformer encoder for edge prediction, it effectively identifies complex error correlations.
>
> References:\
> [1]-[6] The exact references appear in our response to Reviewer q6rR under Weakness 3.\
> [7] DeMarti iOlius et al., "Performance enhancement of surface codes via recursive minimum-weight perfect-match decoding", Phys. Rev. A (2023).
> ## Question 1 - Generalization
> We thank the reviewer. NMWPM demonstrates strong flexibility: new rebuttal tests show it outperforms standard MWPM on the Repetition code under circuit-level noise. Trained on just 9 physical error rate ($p$) points, it seamlessly interpolates across an 18-point range and maintains its edge in strict extrapolation tests outside the training distribution ($p=0.03$ and $p=0.23$). Extended evaluations to $d=14$ confirm scalability, proving the weighting scheme captures geometric correlations rather than memorizing instances. These expanded results (detailed below) will be added to the manuscript.
>
> | **Code ($p$)** | **MWPM** | **NMWPM (Ours)** |
> | :--- | :--- | :--- |
> | Toric $L=8$ (0.030) | 0.0005 | 0.0001 |
> | Toric $L=8$ (0.230) | 0.8324 | 0.7568 |
> | Rotated $L=7$ (0.030) | 0.0015 | 0.0007 |
> | Rotated $L=7$ (0.230) | 0.5048 | 0.4475 |
> ## Weakness & Question 2 - Computational Scalability
> We thank the reviewer for inquiring about computational efficiency. At $L=10$ (depolarizing), generating a training sample averages 0.0158 seconds. Training requires ~700 seconds/epoch, converging within 600 epochs. For inference, NMWPM scales significantly better than QECCT: total runtime increases by only 1.54 ms between $L=6$ and $L=10$, whereas QECCT's latency rises from 7.00 ms to 20.1 ms. We will add this scaling analysis to the updated paper.
>
> | $L$ | QECCT Total | NMWPM GPU | NMWPM CPU | NMWPM Total |
> | :--- | :--- | :--- | :--- | :--- |
> | $6$ | 7.00 ms | 2.23 ms | 0.24 ms | **2.47 ms** |
> | $10$ | 20.1 ms | 2.82 ms | 1.19 ms | **4.01 ms** |
> ## Limitations
> We will revise the manuscript to include a self-contained methodological overview and explicit algorithmic pipeline. To address concerns regarding training costs, we conducted a transfer learning experiment during the rebuttal period: initializing a model for a smaller code size using the learned Transformer weights of a larger model alongside standard random GNN weights yielded fast convergence in just 20 epochs. This demonstrates that the learned features are highly transferable, drastically reducing the computational overhead required for new regimes.

---

> > ### Author Rebuttal · Reviewer_E3CG · 2026-04-05
> >
> > Thanks to the authors for their reply. Partially resolved. But I still have concerns about the computational cost of data generation, model training, inference, generalization capability, and theoretical guarantees as the system size grows, as these are crucial for evaluating whether the approach remains viable at larger scales.

---

> > > ### Author Response · Authors · 2026-04-07
> > >
> > > We appreciate the reviewer's follow-up and the opportunity to further clarify our work. We address the remaining concerns below:
> > > ## Data Generation and Training Cost
> > > Data generation occurs dynamically on the fly during the training loop. Therefore, the ~700 seconds/epoch metric provided in our previous response (using a single L40 GPU) is inclusive of data and label generation times for both graphs (X and Z).
> > > To extend this to larger scales:\
> > > For $L=12$, the time is 800 seconds per epoch, with convergence achieved within 700 epochs\
> > > For $L=14$, the time is 900 seconds per epoch, with convergence achieved within 750 epochs.\
> > > These times can be further reduced via multi-GPU acceleration. In contrast, we could not evaluate QECCT for $L \ge 12$ because it exceeds GPU memory constraints.
> > >
> > > ## Model Scalability
> > > As system size grows, NMWPM's parameter count remains nearly constant, making it highly viable for large-scale. This is a significant advantage over QECCT, whose parameter count grows rapidly:
> > > | **Code Distance ($L$)** | **NMWPM Parameters (Millions)** | **QECCT Parameters (Millions)** |
> > > | :--- | :--- | :--- |
> > > | $L=6$ | 3.99 | 1.90 |
> > > | $L=8$ | 3.99 | 3.42 |
> > > | $L=10$ | 4.00 | 6.64 |
> > > | $L=12$ | 4.02 | 12.50 |
> > >
> > > The stability in NMWPM's parameter count is driven by its hybrid architecture, which learns correlations directly from the sparse syndrome graph. By focusing on this sparse representation, NMWPM avoids the massive parameter growth required by global attention-based models like QECCT, whose complexity and parameter counts increase significantly to handle growing system size.
> > > ## Generalization Capability
> > > To concretely address the concern regarding generalization capability, we present the quantitative results of our transfer learning experiment. We initialized a model for a smaller code size ($L=8$) using the learned Transformer weights of a larger model, alongside standard random initialization for the GNN weights. As shown in the table below, fine-tuning this transferred model for only 20 epochs yields logical error rates that vastly outperform the standard MWPM baseline.
> > >
> > > | Physical Error Rate ($p$) | MWPM | NMWPM (20 Epochs, Transferred)
> > > | :--- | :--- | :--- |
> > > | 0.103 | 0.1333 | 0.0602 |
> > > | 0.129 | 0.2906 | 0.1654 |
> > > | 0.191 | 0.6940 | 0.5752 |
> > >
> > > **Implications for Generalization**
> > >
> > > This rapid convergence provides empirical evidence of cross-scale generalization. If the model merely memorized lattice specific errors, larger model weights would fail on smaller ones. Instead, the strong performance achieved in just 20 epochs shows that the Transformer layers learn size-independent features about the underlying error distributions. This transferability demonstrates that the architecture generalizes across system sizes, while validating a practical method for drastically reducing computational training costs in new regimes.
> > >
> > > Furthermore, to strengthen our claims regarding the model's ability to generalize outside the training distribution, we have extended our extrapolation evaluations to larger system size, $L=10$. As demonstrated in the results below, NMWPM successfully maintains its performance edge over the baseline in these expanded scaling tests.
> > > | **Code ($p$)** | **MWPM** | **NMWPM (Ours)** |
> > > | :--- | :--- | :--- |
> > > | Toric $L=10$ (0.030) | 0.0001 | 0.0000 |
> > > | Toric $L=10$ (0.230) | 0.864 | 0.79 |
> > >
> > > ## Inference Scalability
> > > To directly address the concerns regarding inference as system size grows, we emphasize that NMWPM demonstrates improved scaling behavior compared to baseline neural decoders. When scaling the code distance from $L=6$ to $L=10$, QECCT's inference latency experiences a significant 187% increase (jumping from 7.00 to 20.1 ms). In contrast, NMWPM's total runtime grows by only 62% (an increase of just 1.54 ms). This robust scaling is a direct result of our architectural design. While the GNN efficiently processes the full stabilizer node matrix, the more computationally demanding Transformer encoder predicts weights exclusively for the active edges of the syndrome graph. By restricting the attention mechanism to existing errors rather than applying it across the entire dense lattice, the computational overhead remains small. Consequently, the neural processing time on the GPU increases by merely 0.59 ms between $L=6$ and $L=10$. The remaining workload is handled by the fast, MWPM algorithm on the CPU. This separation ensures that predicting edge weights on the syndrome graph does not introduce a computational bottleneck as the lattice expands.
> > > ## Theoretical Guarantees
> > > Finally, NMWPM preserves the theoretical guarantees of classical MWPM. Because QWP's role is strictly limited to predicting edge weights for the syndrome graph, and the actual decoding is offloaded to the classical MWPM, the decoder is guaranteed to output a valid homological correction. It cannot produce unphysical states, ensuring reliable operation at any scale.

---

### Official Review · Reviewer_ho5Q · 2026-03-12

**Soundness:** 3
**Presentation:** 3
**Significance:** 3
**Originality:** 3
**Overall Recommendation:** 4
**Confidence:** 2

**Summary:**

This paper introduces Neural Minimum Weight Perfect Matching (NMWPM), a novel data-driven decoder used to predict dynamic edge weights in the traditional MWPM algorithms for quantum error correction (QEC). The core contribution is a hybrid architecture, Quantum Weight Predictor (QWP), combining Graph Neural Networks (GNNs) and a global transformer encoder:

- The GNN is used to extract local topological features from the given syndrome graphs.

- The transformer encoder is designed to capture long-range global dependencies on the graphs.

- The authors introduce a procedure to generate ground truth data, reducing sampled physical error configurations to the corresponding equivalence class of error chains.

- They propose a proxy loss function using binary cross-entropy augmented with entropy regularization, overcoming the non-differentiable nature.


The authors conduct experiments on Toric and rotated surface codes, demonstrating the NMWPM reduces the logical error rate and achieves near-optimal error thresholds.

**Compliance With Llm Reviewing Policy:**

Affirmed.

**Final Justification:**

The authors' rebuttal addressed most of my concerns. The additional extrapolation experiments they provided have confirmed the generalizability of their method to some extent. Therefore I raise my recommendation to 4.

**Key Questions For Authors:**

1. Does this method essentially indeed learn the true conditional probability distribution given the syndrome, based on the sampled dataset?

2. In the Transformer encoder processing stage, why does the model utilize directed edge representations instead of undirected edges directly? What is the topological or physical intuition driving this design?

3. A small typo: A missing period at the end of the Abstract.

**Limitations:**

yes

**Strengths And Weaknesses:**

- Instead of adopting an end-to-end design to directly predict the error equivalence class, this model integrates the classical MWPM algorithm with machine learning. By shifting the focus to predicting the edge probabilities of the syndrome graph, the model successfully circumvents the difficulties brought by the inherent discreteness of the problem itself.
- The proxy loss function with an entropy regularization term assists the following classical algorithm. This enforces a sharp dichotomy in predicted probabilities, forces bimodal predictions with high-confidence, and thus reduce the search space for the classical matching algorithm.
- I have concerns regarding the limited generalizability of this study. The model is data-driven, and it learns representations uniquely corresponding to a specific quantum code topology and a predefined noise model, making it difficult to directly transfer a trained model to other topologies.

---

> ### Author Rebuttal · Authors · 2026-03-31
>
> We appreciate the reviewer's insightful review. Below we address the points raised.
> Please note that all new figures are available at the following anonymous link:
> https://anonymous.4open.science/r/ICML2026_Rebuttal_1234-1B20sashdfkk/
> ## Weakness 1 - Generalization
> We thank the reviewer for raising this point. While our model is indeed data-driven, our extensive empirical evaluations demonstrate strong generalizability across multiple dimensions, showing that it does not simply memorize a single setting.\
> **Cross-Topology and Cross-Noise Generalization:** NMWPM successfully decodes both Toric and Rotated Surface codes across multiple noise models (independent and depolarizing). During the rebuttal period, we expanded our evaluation to include the Repetition code under a realistic circuit-level noise model, where it continued to demonstrate performance improvements over standard MWPM.\
> **Code Distance Scaling:** The model generalizes effectively across 8 different code distances. We have now extended this evaluation to larger distance of $d=14$, confirming the model's stability and scalability.\
> **Interpolation and Extrapolation:** NMWPM is highly robust to variations in the physical error rate. The model was trained on 9 physical error probabilities but evaluated on 18 points within that range, demonstrating excellent interpolation. Furthermore, we evaluated the model on the Toric ($d=8$) and Rotated ($d=7$) codes at error rates strictly outside the training distribution ($p=0.03$ and $p=0.23$). As shown in the extrapolation tables below, NMWPM maintained its high performance in both regimes by consistently achieving lower logical error rates than standard MWPM. We will ensure these interpolation, extrapolation, and expanded evaluation results are clearly highlighted in the revised text.
>
> L=8 Toric Depo
> | p | MWPM | NMWPM (Ours) |
> | :--- | :--- | :--- |
> | 0.030 | 0.0005 | 0.0001 |
> | 0.230 | 0.8324 | 0.7568 |
>
> L=7 Rotated Depo:
> | p | MWPM | NMWPM (Ours) |
> | :--- | :--- | :--- |
> | 0.030 | 0.0015 | 0.0007 |
> | 0.230 | 0.5048 | 0.4475 |
>
> ## Question 1 - Conditional Probability Distribution
> We thank the reviewer for this insightful question. Strictly speaking, the model does not learn the exact true conditional probability distribution of the physical errors given the syndrome. Instead, it learns a regularized proxy distribution over edge assignments that is explicitly optimized for the MWPM.
> Because topological codes exhibit degeneracy, the true conditional probability of an individual edge belonging to the physical error chain would often be a soft value (for example, 0.5 when two equivalent paths exist). However, our training objective incorporates an entropy regularization term that deliberately penalizes these soft probabilities. This regularization forces the network to make bimodal, high-confidence predictions (close to 0 or 1). Therefore, rather than performing exact Bayesian probabilistic inference, the model learns an effective heuristic distribution that drastically reduces ambiguity and search space for the matching algorithm as shown in the emprical results.
> ## Question 2 - Directed Edges
> We thank the reviewer for this question. Evaluating both directed permutations ($[x_u, x_v]$ and $[x_v, x_u]$) rather than a single undirected edge maximizes expressivity, preserves spatial symmetry, and aligns with standard graph frameworks.
>
> Constructing a single undirected input forces a choice between two approaches:
> 1. **Symmetrizing Features (Loss of Expressivity):** Pooling node features (for example, $x_u + x_v$) blurs the distinct local topologies of the endpoints, limiting the Transformer's ability to learn complex dependencies.
> 2. **Arbitrary Sorting:** Forcing a consistent node ordering (for example by coordinates) breaks spatial covariance. The network would overfit to an arbitrary boundary rather than learning the translation-invariant physics of the syndrome graph.
>
> **Framework Alignment:** PyTorch Geometric natively represents undirected graphs as pairs of directed edges. Processing these in parallel is computationally efficient and architecturally straightforward. Therefore, retaining both directed edges provides the optimal expressivity, spatial symmetry, and implementation simplicity.
> ## Question 3 - Typo
> We thank the reviewer for bringing this to our attention. We have corrected the missing period at the end of the Abstract in the revised version.

---

> > ### Author Rebuttal · Reviewer_ho5Q · 2026-04-04
> >
> > Thank you for your detailed response to my question and for the additional numerical experiments. My concerns have been largely addressed, and I am therefore pleased to raise my recommendation to 4. However, I believe that the phenomenon in which the trained model performs well under different codes and different error probabilities still requires a more detailed explanation. I hope the authors can elaborate on this point more thoroughly in the revised version.

---

> > > ### Author Response · Authors · 2026-04-05
> > >
> > > We would like to thank you for your positive feedback. We appreciate your thorough review and completely agree with the suggestion. We will make sure to include a deeper discussion regarding the generalization phenomenon in the revised version of the paper.

---

### Official Review · Reviewer_q6rR · 2026-03-12

**Soundness:** 3
**Presentation:** 3
**Significance:** 2
**Originality:** 2
**Overall Recommendation:** 4
**Confidence:** 4

**Summary:**

The paper proposes a neural-guided MWPM decoder that uses a GNN plus a Transformer to predict dynamic edge weights, together with a proxy loss function consisting of binary cross-entropy and regularized entropy. It evaluates toric and rotated surface codes under independent and depolarizing Pauli noise, and compares mainly against MWPM, BPOSD-2, and QECCT. The paper reports theoretical computational complexity of $O(E^2 d_{hidden} + Ed^2_{hidden})$ and parameter efficiency better than QECCT. It reports  thresholds of $17.9\%$ and $10.95\%$, nearing the $18.9\%$ and $11.0\%$ maximum likelihood bounds.

**Compliance With Llm Reviewing Policy:**

Affirmed.

**Final Justification:**

1. good QEC threshold results
2. fair practical implications with short inference time.
3. interesting new design with GNN + Transformer

**Key Questions For Authors:**

1. Can the authors provide inference time or decoding latency benchmarks for their method and compare them against the main baselines?
2. Can the authors provide some evaluation results on some other code families?

**Limitations:**

yes

**Strengths And Weaknesses:**

Strengths:
1. Clear and well-motivated hybrid architecture combining GNN, Transformer, and MWPM.
2. Strong reported threshold performance relative to the paper’s chosen baselines.
3. Good overall exposition, with helpful background and clear figures.
4. Includes theoretical complexity discussion and parameter-count comparison.

Weaknesses:
1. No inference-time or decoding-latency benchmarks, despite runtime being central for practical quantum error correction.
2. Evaluation is limited to toric and rotated surface codes, with no evidence of broader generalization on other code families, such as color codes or repetition codes.
3. Significance is limited due to the large existing literature on neural and non-neural decoders.
4. The architectural complexity and novelty are not well justified for practical quantum error correction.

---

> ### Author Rebuttal · Authors · 2026-03-31
>
> We thank the reviewer for their constructive feedback and address their concerns below.
> Please note that all new figures are available at the following anonymous link:
> https://anonymous.4open.science/r/ICML2026_Rebuttal_1234-1B20sashdfkk/
> ## Weakness & Question 1 - Inference Time
> We thank the reviewer for highlighting this important point. We agree that runtime is a critical metric for practical quantum error correction. To address this, we conducted inference-time benchmark using a single NVIDIA L40 GPU during the rebuttal period. We will add a dedicated discussion regarding this topic, along with the following table.
>
> | $L$ | **QECCT Total** | **NMWPM GPU** | **NMWPM CPU** | **NMWPM Total** |
> | :--- | :--- | :--- | :--- | :--- |
> | $6$ | 7.00 ms | 2.23 ms | 0.24 ms | **2.47 ms** |
> | $10$ | 20.1 ms | 2.82 ms | 1.19 ms | **4.01 ms** |
>
> As shown in the table, NMWPM consistently achieves faster total inference times than QECCT, this advantage becomes more pronounced as the code distance increases. At $L=6$, NMWPM's total runtime is $2.47$ ms compared to $7.00$ ms for QECCT. When scaling to $L=10$, QECCT's runtime increase to $20.1$ ms. In contrast, NMWPM scales much more efficiently, with the total runtime increasing to only $4.01$ ms. This efficiency is achieved because the majority of the computational load is processed on the GPU, which scales well with lattice size, increasing from $2.23$ ms to $2.82$ ms between $L=6$ and $L=10$.
> ## Weakness 2 & Question 2 - Other Code Families
> We thank the reviewer for this suggestion. To evidence broader generalization across code families, we expanded our rebuttal evaluation to the repetition code ($L=3$ and $L=5$) under a realistic circuit-level noise model. These experiments confirm NMWPM adapts effectively beyond Toric and Rotated Surface codes. Demonstrating strong performance on a different topology under complex noise establishes that our hybrid approach is not strictly bound to 2D surface-like spatial structures.
> ## Weakness 3 - Significance
> We thank the reviewer for this feedback. While there is indeed extensive literature on neural and non-neural decoders, the significance of NMWPM lies in effectively bridging these two domains to address their respective limitations.
> Unlike purely neural end-to-end decoders that often struggle with scalability, or standard classical decoders that struggle with complex correlated noise, NMWPM is specifically designed to augment the established MWPM. This hybrid approach yields state-of-the-art results. NMWPM achieves a threshold of 17.9% for the toric code and 17.7% for the rotated surface code. To concretely demonstrate the significance of this performance, the table below highlights the reported thresholds on surface codes compared to recent prominent decoders:
> | **Decoder** | **Threshold Depolarizing Noise ($p_{th}$)** |
> | :--- | :--- |
> | NMWPM | 17.9% |
> | QECCT [1] | 17.8% |
> | Astra (2024) [2] | 17.0% |
> | SU-NetQD (2025) [3] | 16.3% |
> | ML+UF (2022) [4] | 16.2% |
> | BP-OSD2 [5] | 16.0% |
> | MWPM [6] | 16.0% |
>
> We will add this comparative analysis to the final version of the paper.
>
> References:
> [1] Choukroun et al., "Deep quantum error correction", AAAI (2024).\
> [2] Maan et al., "Machine Learning Message-Passing for the Scalable Decoding of QLDPC Codes", (2024).\
> [3] Zhang et al., "Self-attention U-Net decoder for toric codes", Physical Review Applied (2025).\
> [4] Meinerz et al., "Scalable Neural Decoder for Topological Surface Codes", Physical Review Letters (2022).\
> [5] ldpc package (BP-OSD Decoder), our findings.\
> [6] Wang et al., "Threshold error rates for the toric and surface codes", (2009).
>
> ## Weakness 4 - Architectural Complexity and Novelty
> We thank the reviewer for raising the important issue of practicality. NMWPM is designed as a novel middle ground that bridges classical and neural decoding. While standard MWPM is highly scalable, it cannot naturally account for complex noise correlations. Conversely, many neural decoders struggle to scale because they attempt to process the entire physical lattice.
> Our hybrid approach overcomes this by using a neural architecture to learn correlations from the sparse, syndrome graph, while relying on MWPM for the final decoding step. As shown in the latency table above, this efficiency allows NMWPM to consistently outperform global attention-based decoders like QECCT. Furthermore, NMWPM avoids the massive parameter growth seen in global attention models. As demonstrated below, increasing the code distance ($L$) results in only a marginal increase in trainable parameters for NMWPM, whereas QECCT's complexity grows significantly:
>
> | **Code Distance ($L$)** | **NMWPM Parameters (Millions)** | **QECCT Parameters (Millions)** |
> | :--- | :--- | :--- |
> | $L=6$ | 3.99 | 1.90 |
> | $L=8$ | 3.99 | 3.42 |
> | $L=10$ | 4.00 | 6.64 |
> | $L=12$ | 4.02 | 12.50 |
>
> This stability highlights the practical efficiency and scalability of the NMWPM architecture.

---

> > ### Author Rebuttal · Reviewer_q6rR · 2026-04-03
> >
> > I appreciate the author's response in the rebuttal. My questions are mostly resolved. I am happy to change the score as I see fit.

---

> > > ### Author Response · Authors · 2026-04-05
> > >
> > > Thank you very much for your acknowledgment and for taking the time to review our rebuttal. We are glad to hear that our responses have addressed your concerns.
> > >
> > > If you feel it is appropriate, we would greatly appreciate it if you could consider updating your score to reflect your current assessment.
> > >
> > > Thank you again for your time and thoughtful positive feedback.

---

### Official Review · Reviewer_muBL · 2026-03-13

**Soundness:** 3
**Presentation:** 3
**Significance:** 3
**Originality:** 3
**Overall Recommendation:** 4
**Confidence:** 4

**Summary:**

This paper proposes the neural minimum weight perfect matching (NMWPM), a hybrid decoder for topological quantum error-correcting codes that learns syndrome-dependent edge weights for a classical MWPM backend. The method uses a GNN to encode local syndrome structure and a transformer encoder to model global dependencies between candidate error chains, and then converts predicted edge probabilities into MWPM costs. Because MWPM is non-differentiable, the authors train the model with a proxy edge-classification objective based on binary cross-entropy with entropy regularization, together with a heuristic procedure for constructing ground-truth matchings from simulated error configurations. Experiments on toric and rotated surface codes under independent and depolarizing noise show consistent logical error rate improvements over MWPM, BPOSD-2, and QECCT, with reported thresholds of 17.9% on toric depolarizing noise, 10.95% on toric independent noise, and 17.7% on rotated-surface depolarizing noise. Overall, the paper argues that learned, shot-dependent priors can substantially strengthen classical matching-based decoders without discarding their algorithmic structure.

**Compliance With Llm Reviewing Policy:**

Affirmed.

**Final Justification:**

While the rebuttal effectively clarified boundary handling and circuit-level performance, the reliance on a heuristic labeling strategy under high degeneracy remains a conceptual limitation for capturing the true conditional probability distribution. Overall, the work offers a meaningful contribution to hybrid quantum decoding, proving that shot-dependent, learned weights can materially enhance the robustness of established combinatorial solvers for topological codes.

**Key Questions For Authors:**

1. How reliable is the heuristic ground-truth construction used for supervision in the presence of degeneracy?
It would be helpful if the authors could report the fraction of samples that require permutation search or timed brute-force fallback, the timeout budget used for the fallback, and how cases are handled when a valid correction cannot be found within the allowed time. It would also be useful to explain how often multiple valid matchings exist for the same syndrome, and whether any residual label noise remains even after the fallback procedure. Since the learning objective depends entirely on these labels, clarifying this issue would significantly increase my confidence in the soundness of the paper.

2. Do the main conclusions continue to hold in more realistic settings, such as larger code distances, repeated syndrome rounds, or circuit-level / measurement noise?
Even a limited additional experiment, or a careful explanation of why this is left to future work, would be helpful. If it can be shown that the observed improvements are not confined to the code-capacity regime, my evaluation would become more positive.

3. Could the authors clarify how defect-to-boundary edges are handled at inference time for the rotated surface code?
The paper explicitly states that a virtual node is introduced for boundary handling during ground-truth construction when a cluster contains an odd number of endpoints. In contrast, the main decoding pipeline is described as operating on a complete graph over active defects. It is therefore unclear whether boundary nodes/edges are also included in the learned graph during inference and assigned neural weights, or whether boundary edges are handled separately by the MWPM backend using fixed costs. Clarifying this point would improve both the reproducibility of the method and my confidence in the rotated-surface-code results.

4. Could the authors provide either a comparison with, or a more direct discussion of, representative MWPM-enhanced or correlation-aware decoders?
Since the main contribution of this paper is to improve MWPM rather than replace it, the novelty and practical value of the work could be assessed much more accurately if the paper included either performance comparisons with representative correlated/pipelined MWPM variants or with prior work that explicitly reweights the matching graph, or at least a clearer discussion positioning the method relative to such approaches.

**Limitations:**

The paper does discuss potential negative societal impact through its dual-use connection to cryptography. However, the discussion of technical limitations is less complete. In particular, the paper would benefit from a more explicit discussion of its restricted evaluation setting (independent/depolarizing noise, moderate code distances), the reliance on heuristic ground-truth construction under degeneracy, the lack of runtime/latency benchmarks, and the unclear treatment of boundary handling for the rotated surface code.

**Strengths And Weaknesses:**

• The division of roles between the GNN, which processes local geometric/topological structure, and the transformer, which infers global relationships among candidate edges, is well coordinated.

• However, the most important technical weakness is that the learning supervision relies on a heuristic ground-truth construction in a setting with degeneracy. The paper does not quantify the ambiguity of these labels, nor does it report the fraction of samples that require permutation search or timed brute-force fallback, or explain how cases are handled when the timed search fails to find a valid correction within the allotted time. Given that the model’s learning objective depends entirely on these labels, this is important.

• In addition, the current experiments are limited to independent noise, depolarizing noise, and moderate code distances, and therefore do not yet demonstrate whether the same advantages persist in more realistic repeated-syndrome settings or under circuit-level noise.

• The trial of the ablation study is good but it shows that each individual component does not have a significant contribution.

• The paper is generally easy to read and well structured. However, some implementation details are not specified clearly enough from a reproducibility standpoint. In particular, it remains unclear how defect-to-boundary connections are handled for the rotated surface code. The paper explicitly states that virtual nodes are introduced for boundary handling during ground-truth construction, yet the main decoding pipeline is described as operating on a complete graph over active defects. As a result, it is not clear whether boundary edges are also included in the learned graph at inference time and assigned predicted weights, or whether they are handled separately by the MWPM backend using fixed costs.

• In addition, the positioning with respect to related work could be strengthened. While the current comparisons are adequate with respect to standard MWPM and a relatively broad set of neural baselines, the paper’s position would be clearer if it included more direct conceptual and empirical comparisons with correlation-aware MWPM methods or MWPM-augmented decoders.

• Improving practical decoding for topological quantum codes is a highly significant topic, and the methodological idea of assigning learned dynamic priors to a classical combinatorial decoder has the potential to be useful beyond this specific architecture.

• The significance of the work would be considerably stronger if the authors provided evidence that the advantages persist under more realistic noise models and operational constraints.

• I view the originality as a meaningful combinational novelty rather than the introduction of a completely new paradigm. The paper does not propose an entirely new decoder family, but it combines learned edge reweighting, GNN-based local processing, transformer-based global reasoning, and classical MWPM inference in a nontrivial and well-coordinated way. The key novelty lies in formulating this hybrid pipeline and demonstrating that shot-dependent learned weights can materially improve a matching-based decoder in practice.

---

> ### Author Rebuttal · Authors · 2026-03-31
>
> We thank the reviewer for their thoughtful feedback and for recognizing the meaningful combinational novelty and well-coordinated design of our NMWPM architecture. We will now address every weakness and question in detail:
> Please note that all new figures are available at the following anonymous link:
> https://anonymous.4open.science/r/ICML2026_Rebuttal_1234-1B20sashdfkk/
> ## Weakness and Question 1 - Heuristic Ground Truth
> We thank the reviewer and will add these crucial ground-truth generation details to the revised manuscript.\
> **Search Fraction & Timeout:** For the Toric code ($L=8$) under depolarizing noise, only 0.6% of the dataset requires brute-force search. We enforce a strict 10-second timeout per sample to balance generation time and data retention.\
> **Failed Searches & Label Noise:** Samples exceeding the 10-second limit are discarded, amounting to only 0.2% of $L=8$ samples. By exclusively training on verified matchings, we eliminate residual label noise. This filtering applies strictly to training data generation, absolutely no samples are discarded during inference or evaluation.\
> **Ambiguity & Degeneracy:** Although multiple valid matchings exist per syndrome, our deterministic heuristic assigns the first valid matching found as the ground truth. We do not exhaustively quantify total label ambiguity. This early stopping ensures a clean training signal, saves computation, and naturally produces non-trivial labels. By deviating from standard MWPM shortest-path solutions, these labels teach the model to succeed where standard MWPM typically fails.
> ## Weaknesses 2, 6 and Question 2 - Realistic Settings and Circuit-Level Noise
> We appreciate the reviewer's suggestion to evaluate our approach in more realistic environment. To address this, we conducted additional experiments during the rebuttal period. Under circuit-level noise with $d$ measurement rounds, we evaluated our model on the repetition code ($d=3, 5$) and a rotated X-memory experiment ($d=3$). In all of these settings, our method consistently demonstrates an improvement over MWPM. Scaling: We evaluated the Toric code under depolarizing noise at $d=14$, confirming our method's advantages persist at scale.
> ## Weakness 3 - Ablation
> We appreciate the reviewer's observation. While removing some individual features yields minor performance drops, this actually reflects a robust architecture where components partially compensate for one another. The model's overall performance relies on their combined effect. Importantly, altering the core architecture does cause a major performance drop. As shown in our ablation study, replacing the Transformer encoder with a standard MLP results in significant degradation. This confirms that the attention mechanism itself is essential.
> ## Weakness 4 & Question 3 - Boundary Handling
> We thank the reviewer for highlighting this, clarifying our boundary treatment is essential for reproducibility. The virtual boundary node is incorporated into the learned graph. We augment the node feature matrix with an additional row representing this virtual node, connecting it to all active defect nodes to maintain a complete graph. This allows the model to capture spatial correlations (bulk and boundary errors). Consequently, QWP predicts the probability of an error chain connecting each active defect to the boundary, assigning a learned weight to every defect-to-boundary edge. We will detail this mechanism in an appendix in the revised manuscript.
> ## Weakness 5 & Question 4 - Comparison with Correlation-Aware MWPM
> We thank the reviewer for this suggestion. To accurately assess practical value, we implemented direct performance comparisons with a correlated MWPM decoder on both Toric and Rotated Surface codes under depolarizing noise. In all evaluations, NMWPM consistently outperformed the correlated MWPM baseline. Furthermore, against recursive MWPM [1] on the rotated surface code (depolarizing), NMWPM achieves a significantly higher threshold ($17.7\%$) compared to the recursive baseline ($16.5\%$). We will include these experiments in the manuscript.
>
> [1] DeMarti iOlius et al., "Performance enhancement of surface codes via recursive minimum-weight perfect-match decoding", Phys. Rev. A (2023).
> ## Limitations
> We agree that discussing technical limitations strengthens the paper. We address these concerns above and provide the requested inference-time benchmarks against QECCT below:
>
> | $L$ | **QECCT Total** | **NMWPM GPU** | **NMWPM CPU** | **NMWPM Total** |
> | :--- | :--- | :--- | :--- | :--- |
> | $6$ | 4.10 ms | 2.23 ms | 0.24 ms | **2.47 ms** |
> | $10$ | 20.1 ms | 2.82 ms | 1.19 ms | **4.01 ms** |
>
> NMWPM scales efficiently: at $L=10$, QECCT latency reaches 20.1 ms, whereas NMWPM requires only 4.01 ms. This efficiency is driven by highly stable GPU processing loads (2.23 to 2.82 ms). We will expand the Limitations section and Appendix in the revised manuscript to incorporate these technical discussions and new benchmarks.

---

> > ### Author Rebuttal · Reviewer_muBL · 2026-04-03
> >
> > Thank you for the comprehensive rebuttal and the additional experimental data.
> > Most of my concerns are properly addressed.
> > However, the reliance on a heuristic "first valid matching" for ground-truth labels still sidesteps the deeper challenge of learning the true conditional probability distribution in the presence of high degeneracy.
> > Because the circuit-level evaluations are still relatively limited to smaller distances and simpler topologies, there remains some uncertainty regarding the decoder's performance in full space-time decoding regimes.
> > Consequently, I maintain my recommendation as the work offers a technically sound and meaningful integration of learned priors with classical algorithmic structures.

---

### Decision · Program_Chairs · 2026-04-30

**Decision:**

Accept (regular)

**Comment:**

The paper presents an interesting hybrid decoding framework that combines learned edge-weight prediction with MWPM, and I agree with the positive reviewers that the overall design is technically thoughtful and supported by promising threshold results. The use of GNNs and Transformers within this structured decoding pipeline is a strength, and the proxy loss used to train through a non-differentiable solver is also a meaningful component of the contribution.

That said, I still have reservations about the empirical completeness of the work, particularly with respect to scalability. In particular, the current evidence is not yet sufficient to establish that the approach remains effective and practical at larger code distances or under broader and more standard evaluation settings, such as circuit-level noise models. Stronger comparisons against state-of-the-art neural decoder baselines, as well as evaluations under more widely used noise models, would have strengthened the paper considerably.

I explicitly raised these concerns during the reviewer discussion and asked whether the more positive reviewers believed they had been adequately resolved, but no further clarification was provided. Nevertheless, on balance, I find the proposed method technically interesting, the empirical results sufficiently promising, and the overall contribution strong enough to justify acceptance, albeit with some caution.